# Multivariable Iterative Learning Control Design for Precision Control of Flexible Feed Drives

**DOI:** 10.3390/s24113536

**Published:** 2024-05-30

**Authors:** Yulin Wang, Tesheng Hsiao

**Affiliations:** 1Institute of Electrical and Control Engineering, National Yang Ming Chiao Tung University, Hsinchu 30010, Taiwan; yu.linwang106@gmail.com; 2School of Mechanical Engineering, Guangdong Ocean University, Zhanjiang 524088, China

**Keywords:** iterative learning control, flexible feed drives, multivariable control, norm-optimal

## Abstract

Advancements in machining technology demand higher speeds and precision, necessitating improved control systems in equipment like CNC machine tools. Due to lead errors, structural vibrations, and thermal deformation, commercial CNC controllers commonly use rotary encoders in the motor side to close the position loop, aiming to prevent insufficient stability and premature wear and damage of components. This paper introduces a multivariable iterative learning control (MILC) method tailored for flexible feed drive systems, focusing on enhancing dynamic positioning accuracy. The MILC employs error data from both the motor and table sides, enhancing precision by injecting compensation commands into both the reference trajectory and control command through a norm-optimization process. This method effectively mitigates conflicts between feedback control (FBC) and traditional iterative learning control (ILC) in flexible structures, achieving smaller tracking errors in the table side. The performance and efficacy of the MILC system are experimentally validated on an industrial biaxial CNC machine tool, demonstrating its potential for precision control in modern machining equipment.

## 1. Introduction

In advanced manufacturing equipment such as manipulators [1], CNC machine tools [2], 3D printers [3], and lithography machines [4], the throughput and accuracy of workpieces are significantly affected by the dynamic positioning accuracy of their feed drive systems. In industrial applications, electromechanical systems are inevitably affected by the structural dynamics of the feed drives and uncertain disturbance, significantly limiting the achievable accuracy of motion control systems. For precision motion systems, the ultimate goal is to use high-performance control algorithms to suppress various disturbances in diverse operational environments. Disturbances affecting control accuracy are typically divided into two main categories: internal and external disturbances.

For CNC machine tool motion systems, cutting forces between the workpiece and the tool are common forms of external disturbances. High-gain feedback controllers are typically utilized to mitigate these disturbances induced by the process. To further enhance performance, more advanced motion controllers have been considered. These include pole placement [5], loop shaping [6], and $H_{\infty }$ synthesis strategies [7]. Additionally, feedback control (FBC), using nonlinear control techniques like sliding mode control [8] and model predictive control [9], was adopted to enhance the dynamic positioning accuracy of motion systems.

On the other hand, the presence of internal disturbances, such as lead errors, thermal errors, and weight deformation errors in ball screws, also limits commercial CNC machines to closing the position loop using rotary encoder feedback. These disturbances can lead to premature wear and damage to mechanical components when a linear encoder is used to close the position loop [10]. Recent studies have illustrated that finite element modeling (FEM) methods can adeptly mitigate these inaccuracies [11]. However, the pre-calibration process is notably time-consuming and requires substantial prior knowledge, obstructing its widespread adoption in engineering applications. Additionally, due to the inherent flexibility of the feed drive system, the accuracy at the motor side does not directly correspond to the required precision at the table side, which is of primary interest. In light of this, the present study proposes a strategy that iteratively refines the reference trajectory commands based on the observed results from the linear encoder on the table side.

Residual vibrations pose another significant challenge to the dynamic accuracy of precision motion stages, primarily stemming from the structural dynamics like resonance damping in flexible feed drive components. These vibrations are induced by inertial forces generated during high-speed and high-acceleration trajectories. However, FBC systems, despite incorporating advanced algorithms, inherently experience delays and transient responses to changes in reference inputs and disturbances. Conversely, a feedforward control signal can be designed to preemptively target future reference commands and disturbances, utilizing an understanding of the system’s dynamics and the information at hand. Techniques like notch filtering [12] and input shaping [13] are employed to modify the spectral energy around the resonance frequency in trajectory commands, aiming to reduce vibrations induced by the trajectory. Yet, these approaches may introduce additional delays and can distort trajectory commands, which can compromise the accuracy of contour tracking in multi-axis motion. Furthermore, plant inversion methods such as Zero Phase Error Tracking Control (ZPETC) [14], Zero Magnitude Error Tracking Control (ZMETC), or Nonminimum-Phase Zero Ignore (NPZ-Ignore) [15] can also be used for pre-filtering trajectory commands to prevent vibrations and improve trajectory tracking accuracy. Nonetheless, the effectiveness of these feedforward filtering techniques is dependent on the precise inversion of the model, necessitating an accurate identification of the system’s dynamic model.

Iterative Learning Control (ILC) is a feedforward control strategy designed to enhance the performance of systems carrying out repetitive tasks [9,16,17]. This method utilizes error information collected from previous tasks to enhance the performance of current tasks, effectively suppressing the repetitive disturbances mentioned above. The underlying assumptions of ILC is that the reference signal is repetitive. Each repetition is referred to as an iteration, and the initial conditions (ICs) for each iteration are identical.

Notice that these assumptions are not restrictive, because repetitive machining is very common in batch manufacturing. ILC can be effectively used to synthesize error compensation strategies for such scenarios. It is usually integrated into an existing FBC system since ILC itself is incapable of stabilizing an unstable system or compensating for non-repetitive disturbances. 

In the literature, there are several mainstream frameworks for designing ILC systems, including model-based ILC, data-driven ILC, and adaptive ILC. Model-based ILC generates control compensation using a known system model and previous error data. This strategy is recognized for its simplicity and ease of implementation, featuring methods like frequency-domain inversion-based and time-domain norm-optimal methods [17]. This technique is widely used in manufacturing systems with linear models, such as CNC machines and wafer stages. However, there is still room for improvement, including the study of flexible structures in this paper. Data-driven ILC [18,19,20,21] parameterizes the learning feedforward by introducing basis functions and iteratively obtaining optimal parameters related to the model. This ensures performance under varying reference, but it has limited ability to counter disturbances. Adaptive ILC [22,23,24] employs Lyapunov-like methods to design learning rules and a parametric updating law to estimate system uncertainties. This addresses uncertainties and disturbances in nonlinear systems. Over the past few years, the ILC field has witnessed significant advancement. For instance, Zhang et al. [25] proposed a data-driven ILC combined with predictive control to address the multivariable tracking problem with actuator constraints. This approach transforms the problem into an iteration-varying quadratic programming problem using only the system’s I/O data. Chi et al. [26] developed a data-driven indirect ILC method for repetitive nonlinear systems. They introduced an adaptive iterative learning rule to update the gains of the indirect iterative learning law for set-points. Li et al. [27] presented an iterative learning-based predictive control method for asynchronous switching of multiphase batch processes with complex characteristics. This method was implemented within the framework of a two-dimensional system.

However, these ILCs are designed for minimizing a single variable, and thus, the majority of existing ILC structures are only suitable for rigid body systems, where the motor position and the table position are identical. A straightforward approach to mitigate table-side errors while minimizing changes to the current rotary encoder feedback control structure is to integrate an additional ILC based on linear encoder measurements. However, conflicts between FBC and ILC might arise due to inconsistency between the motor side and table side errors. Therefore, appropriate modifications to traditional ILC architectures become necessary. Despite this, to the best of the authors’ knowledge, only a handful of studies concentrated on ILC design for flexible structures.

Chen et al. [28] proposed a dual-stage ILC for robots with joint elasticity to address model mismatch, but tuning the dual gains could be tedious, and the learning process must be reset whenever the gains are adjusted. Wang et al. [29] introduced a robust $H_{\infty }$ synthesis ILC, which integrated torque and motor reference learning to improve error convergence and overcome the limitations of [28]. However, these approaches require more intricate models, making the design process more complex. J. Wallen et al. [30] designed an observer-based ILC, where table side displacements were inferred from real-time measurements of the motor side displacements, and the standard ILC directly updated the compensation command. Nonetheless, J. Bolder et al. [31] pointed out that this method could result in internal instability if the ILC acted directly on performance variables (i.e., the table side error) while the feedback controller acted on measurement variables (i.e., the motor side error). Dumanli et al. [32] developed multiple prefilters based on flexible dynamic systems to iteratively correct reference inputs and control compensation signals, ultimately reducing errors at both the motor and table side. This approach involves designing each filter separately, leading to a substantial design workload. Overall, these works highlight the importance of well-coordinated learning in feedforward and FBC to achieve high-precision motion.

It has been observed that prefiltering trajectory commands for FBC could overcome the contradictions between FBC and ILC [31]. However, this requires an accurate model, and robustness to disturbances is compromised. Motivated by the concept of prefiltering trajectory commands, this paper introduces a multivariable ILC (MILC) method. This aims to reconcile the conflicts between FBC and ILC, achieving precise table side trajectory tracking in flexible drive systems. MILC constructs an error matrix to correct the trajectory command and the control inputs, allowing for simultaneous suppression of errors in both FBC and ILC. Then we carry out experiments to verify the effectiveness of MILC in a biaxial motion stage.

The primary contribution of this paper is the development of a multivariable ILC (MILC) framework that combines a two-degrees-of-freedom (DOF) architecture with reference shaping and feedforward compensation. This framework aims to enable simultaneous suppression of errors in both the motor side and the table side for systems with flexible feed drives. The specific contributions of this paper are outlined as follows:

C1: A thorough analysis of prevalent ILCs in controlling flexible structures, highlighting the importance of reference shaping for enhancing ILC performance.

C2: The development of a two-DOF-based MILC framework that effectively reduces feedback errors while boosting ILC performance.

C3: Experimental validation of the MILC approach on a motion system, demonstrating its effectiveness.

This research underlines the need for careful consideration of the balance between ILC and FBC objectives when applying ILC to motion systems with flexible feed drives. The remainder of this paper is organized as follows. Section 2 introduces the notation and foundational concepts of model-based ILC. The problem under consideration is formulated in Section 3. Section 4 details a combined reference shaping and control feedforward MILC architecture designed for flexible structure control, including stability analysis and considerations for implementation. Section 5 validates the effectiveness of the proposed method by experiments. Lastly, conclusions are drawn in Section 6.

The following conventions of notations are adopted in this paper. A discrete-time single-input, single-output (SISO) linear time-invariant (LTI) system P is expressed as
(1)$$P=C{\left(zI-A\right)}^{-1}B+D\begin{array}{c} \underline{\underline{z}} \end{array}\left[\begin{array}{cc} A & B\\ C & D \end{array}\right]$$
where z denotes the complex variable of the *z*-transform for a discrete-time system, and *A*, *B*, *C*, and *D* are matrices of the state space.

Let ${\left\|x\right\|}_{M}∶=x^{T}Mx$, where $x\in R^{n}$ and $M\in R^{n\times n}$ is positive definite (M≻0), i.e., $x^{T}Mx>0,\forall x\neq 0$. It is positive semi-definite (M≽0) if and only if $x^{T}Mx\geq 0,\forall x$.

In this paper, the design of ILC is approached through the lifted form, which depicts the input and output of a SISO and LTI system P in the following manner:(2)$$\underset{\underline{y}}{\underbrace{\left[\begin{array}{c} y[0]\\ y[1]\\ \vdots \\ y[N-1] \end{array}\right]}}=\underset{\underline{P}}{\underbrace{\left[\begin{array}{cccc} D & 0 & \cdots & 0\\ h[1] & D & \cdots & 0\\ \vdots & \vdots & \ddots & \vdots \\ h[N-1] & h[N-2] & \cdots & D \end{array}\right]}}\underset{\underline{u}}{\underbrace{\left[\begin{array}{c} u[0]\\ u[1]\\ \vdots \\ u[N-1] \end{array}\right]}}$$
where h is the impulse response of P, $h\left[k\right]=CA^{k-1}B$, k=1,2,…, and N is the length of each iteration. The underline denotes the respective lifted matrix form.

## 2. ILC Preliminaries 

The MILC is a method that evolves from the model-based ILC. This section analyzes the principles of the model-based ILC and the implementation methods of norm-optimal calculations.

The standard ILC setup can be depicted in Figure 1. In this configuration, $r_{d}$ represents the desired reference input; $y_{j}^{m}$ denotes the system trajectory output of the j-th iteration, with the subscript j indicating the iteration number and the superscript m indicating the motor side; $e_{j}$ represents the motion error of the *j*-th iteration; and $u_{j}$ signifies the ILC compensation signal for the same iteration. $P_{M}$ stands for the motor side control plant, which includes actuators, mechanics, and sensors, while $C_{P}$ is a feedback controller. Additionally, a subscript of 0 indicates that the signal originates from the initial iteration prior to the execution of ILC.

The feedback error for the j-th iteration can be expressed as follows:(3)$$e_{j}=r_{d}-y_{j}^{m}=Sr_{d}-Hu_{j}$$
where $S={\left(1+C_{P}P_{M}\right)}^{-1}$ represents the sensitivity function, and $H=SP_{M}$ is the learning model that maps $u_{j}$ (the learning input) to $y_{j}^{m}$ (the desired output). When ILC feedforward is not applied, as in the initial trial, where j=0 and $u_{0}=0$, the feedback error is expressed as $e_{0}=Sr_{d}$. This error can be minimized by reducing the sensitivity function S at frequencies where the power spectrum of $r_{d}$ and disturbances is significant, while also maintaining the stability of the closed-loop system. To achieve this, the paper utilizes a traditional proportional-integral-derivative (PID) type feedback controller.

The ILC compensation signal is calculated as follows:(4)$${\underline{u}}_{j+1}=\underline{L_{u}}{\underline{u}}_{j}+\underline{L_{e}}{\underline{e}}_{j}$$
where $\underline{L_{u}}$ and $\underline{L_{e}}$ denote the learning functions [17,33], which are determined by minimizing the criterion J presented in Definition 1.

**Definition** **1.***(Performance criterion). A commonly used performance criterion for norm-optimal ILC (NOILC) is defined as follows [14]:*(5)$$J\left({\underline{u}}_{j+1},{\underline{e}}_{j+1}\right)={\left\|{\underline{e}}_{j+1}\right\|}_{W_{q}}^{2}+{\left\|{\underline{u}}_{j+1}-{\underline{u}}_{j}\right\|}_{W_{r}}^{2}+{\left\|{\underline{u}}_{j+1}\right\|}_{W_{s}}^{2}$$*with (*$W_{q},\;W_{r},\;W_{s}$*) semi-positive definite weighting matrices, which are employed to impose penalties on* 
$e_{j+1}$*, *
$u_{j+1}-u_{j}$
*, and *
$u_{j+1}$
*, respectively. A smaller* 
$W_{r}$ *results in a larger* 
$u_{j+1}-u_{j}$
*, indicating increased sensitivity of the ILC output to the error information from the previous iteration. As a result, the ILC becomes more vulnerable to non-repetitive disturbances. Conversely, to enhance robust monotonic convergence, a common strategy is to increase* 
$W_{s}$
*, which, however, slows down the error convergence* *[34,35]. Thus, the selection of these weighting matrices involves balancing robustness with performance.**Since* 
${\underline{e}}_{j+1}$ *is affine with respect to *
${\underline{u}}_{j+1}$
*, and* 
$J\left({\underline{u}}_{j+1},{\underline{e}}_{j+1}\right)$
* is a convex quadratic function of * 
${\underline{u}}_{j+1}$
*, the optimal feedforward control compensation* 
${\underline{u}}_{j+1}$ *can be analytically derived by setting the derivative of the cost function to zero:*
(6)$$\frac{dJ\left({\underline{u}}_{j+1},{\underline{e}}_{j+1}\right)}{d{\underline{u}}_{j+1}}=0$$
*thereby identifying the optimal compensation* 
${\underline{u}}_{j+1}^{*}$ 
*for the next iteration, as described in [17] and specified by Theorem 1.*

**Theorem** **1.***(Minimizer of the Performance Criterion in Definition 1) With the model* 
H 
*and data from the previous iteration, including* 
${\underline{u}}_{j}$
*,  *
${\underline{e}}_{j}$
*, and* 
$\underline{r_{d}}$
*, the optimal* 
${\underline{u}}_{j+1}$ *that minimizes the NOILC performance criterion in Definition 1 is:*
(7)$$\underline{L_{u}}={\left({\underline{H}}^{T}W_{q}\underline{H}+W_{s}+W_{r}\right)}^{-1}\left({\underline{H}}^{T}W_{q}\underline{H}+W_{r}\right)$$
(8)$$\underline{L_{e}}={\left({\underline{H}}^{T}W_{q}\underline{H}+W_{s}+W_{r}\right)}^{-1}{\underline{H}}^{T}W_{q}$$

**Proof of Theorem** **1.**By rearranging ${\underline{e}}_{j}$ and ${\underline{e}}_{j+1}$ in (3), and subsequently eliminating $\underline{r_{d}}$, we can derive the error dynamics in the iteration domain as follows:(9)$${\underline{e}}_{j+1}={\underline{e}}_{j}+\underline{H}({\underline{u}}_{j}-{\underline{u}}_{j+1})$$Substitute (9) into (5) and solve (6) to obtain (7) and (8). □

From Theorem 1, it is evident that NOILC is capable of handling both multivariable scenarios and noncausal computation, the latter of which enables it to effectively deal with non-minimum phase systems. However, calculation of the N×N matrix inversion in Equations (7) and (8) is computationally demanding. Consequently, J. van Zundert introduced a more efficient NOILC algorithm [36] that solves Riccati equations to decrease the computational load. For details on this algorithm, please see Appendix A.

## 3. Flexible Feed Drive and Problem Formulation

This section begins by introducing the fundamental two-mass model for flexible ball screw feed drives, which aids in understanding our control objectives. It then proceeds to outline the limitations associated with the direct application of standard ILC to motion systems with flexible drives and feedback from the table side position, which is referred to as Direct ILC (DILC) in this paper.

### 3.1. Dynamics of Flexible Ball Screw Drive

Presently, ball screw (BS) feed drives are the predominant choice in most industrial feed drive systems. A typical ball screw feed drive in CNC machine tools and its corresponding two-mass model are illustrated in Figure 2a,b [25]. Due to the flexible coupling and the long shaft between the motor and the table, this mechanism can be modeled as a flexible two-mass system [37]. The achievable motion accuracy of this system is constrained by structural vibrations. In the diagram, $m_{M}$ and $m_{T}$ represent the mass on the motor side and the table side, respectively. The control input is denoted as u, while $c_{s}$ and $k_{s}$ represent the viscous damping and the finite stiffness coefficients between the two masses. $c_{M}$ and $c_{T}$ signify the viscous friction arising from the guideway and bearing system.

The differential equations governing the behavior of this flexible BS drive are formulated as follows:(10)$$m_{M}{\ddot{y}}^{t}=-c_{M}{\dot{y}}^{m}-k_{s}\left(y^{m}-y^{t}\right)-c_{s}\left({\dot{y}}^{m}-{\dot{y}}^{t}\right)+u\;m_{T}{\ddot{y}}^{m}=-c_{T}{\dot{y}}^{t}+k_{s}\left(y^{m}-y^{t}\right)+c_{s}\left({\dot{y}}^{m}-{\dot{y}}^{t}\right)$$

The aforementioned equations can be translated into the Laplace(s) domain as follows:(11)$$P_{M}\left(s\right)=\frac{m_{T}s^{2}+\left(c_{s}+c_{T}\right)s+k_{s}}{q_{1}s^{4}+q_{2}s^{3}+q_{3}s^{2}+q_{4}s}$$
(12)PZs=PT(s)PM(s)=css+ksmTs2+cT+cMs+ks
where $P_{M}\left(s\right)$ and $P_{T}\left(s\right)$ represent the transfer functions (TF) from the control input u to the motor side and table side position, $y^{m}$ and $y^{t}$, respectively. Then $P_{z}(s)$ defined in (12) is the TF from the motor side position to the table side position. In addition, $q_{1}=m_{M}m_{T}$, $q_{2}=\left(m_{T}+m_{M}\right)c_{s}+m_{M}c_{T}+m_{T}c_{M}$, $q_{3}=\left(m_{T}+m_{M}\right)k_{s}+c_{M}c_{T}+\left(c_{M}+c_{T}\right)c_{s}$, and $q_{4}=\left(c_{M}+c_{T}\right)k_{s}$.

In the case of collocated control, where the sensor and the actuator are located in close proximity, FBC is designed based on $P_{M}(s)$. This setup leads to an antiresonance vibration mode at $\omega_{AR}=\sqrt{k_{s}/m_{T}}$, as shown in Figure 2d. On the other hand, in the non-collocated control strategy, where the feedback signal is taken from the table side ($P_{T}(s)$), the system exhibits the natural resonance vibration mode at $\omega_{R}=\sqrt{k_{s}(m_{T}+m_{M})/m_{M}m_{T}}$, as illustrated in Figure 2e.

Both the resonance and anti-resonance vibration modes can cause relative motion between the motor and the table side, and adversely affect the positioning accuracy of the table. This fact highlights that it is necessary to control both the motor side and the table side simultaneously. A majority of industrial servo closed-loop control systems primarily employ the collocated control approach to guarantee larger stability margins [38]. Additionally, mechanical manufacturing errors such as lead errors, and thermal deformation of the BS, can cause premature wear and damage if a non-collocated control strategy is directly utilized.

Therefore, building upon the arguments presented in Section 1, we propose an MILC method based on collocated control in Section 4. This approach involves iterative tuning to both the control compensation and the reference input, aiming to simultaneously suppress position errors on both the motor side and the table side. To demonstrate the effectiveness of MILC, the following subsection presents the theoretical analysis and proof of traditional ILC directly applied to flexible structures.

### 3.2. Applying Standard ILC to the Flexible Structure

As illustrated in Figure 3 [30], the traditional ILC, when directly applied to flexible structures, is referred to as DILC in this paper. Within the DILC framework, the feedback controller regulates the motor side error, i.e., $e_{j}^{c1}=r_{d}-y_{j}^{m}$, while the ILC is designed to control the table side error, i.e., $e_{j}^{z1}=r_{d}-y_{j}^{t}$.

Instead of analyzing the performance of DILC in the time domain, this paper leverages the frequency domain. Frequency domain analysis, a well-established tool for infinite-time horizon, is also widely utilized in ILC systems [17], where the repetitive signals have finite lengths. Employing a frequency domain approach sheds light on the system’s performance across different frequencies, making the interaction between FBC and DILC more comprehensible.

Define the total control input to the plant as $v_{j}=u_{j}+u_{j}^{c}=u_{j}+C_{P}(r_{d}-P_{M}v_{j})$; then
(13)$$v_{j}=Su_{j}+SC_{P}r_{d}$$

The error on the table side is expressed as:(14)$$\begin{array}{ll} e_{j}^{z1}=r_{d}-y_{j}^{t} & =r_{d}-P_{Z}P_{M}v_{j}=\left(1-P_{Z}P_{M}SC_{P}\right)r_{d}-P_{Z}P_{M}Su_{j}\\ & =\left(1-P_{Z}\left(1-S\right)\right)r_{d}-P_{Z}P_{M}Su_{j} \end{array}$$

By comparing (3) and (14), it is observed that the learning model for DILC is represented by $H_{D}=SP_{M}P_{Z}$. On the assumption that the ILC stability condition, as detailed in [17], is satisfied, we proceed in our analysis to the steady-state behavior (i.e., j→∞) of DILC. The steady state signals of $u_{j}$, $u_{j}^{c}$, $e_{j}^{z1},$ and $e_{j}^{c1}$ are henceforth denoted as $u_{\infty }$, $u_{\infty }^{c}$, $e_{\infty }^{z1},$ and $e_{\infty }^{c1}$, respectively. Employing Equation (4) as a basis, we derive the following equation:(15)$$u_{\infty }={\left(1-L_{u}\right)}^{-1}L_{e}e_{\infty }^{z1}$$

By substituting (15) into (14), we obtain
(16)$$\begin{array}{ll} e_{\infty }^{z1} & =\left(1-P_{Z}\left(1-S\right)\right)r_{d}-P_{Z}P_{M}Su_{\infty }\\ & =\left(1-L_{u}\right){\left(1-L_{u}+P_{Z}P_{M}SL_{e}\right)}^{-1}\left(1-P_{Z}\left(1-S\right)\right)r_{d} \end{array}$$

Consequently, the steady-state output of DILC is given by
(17)$$u_{\infty }=L_{e}{\left(1-L_{u}+P_{Z}P_{M}SL_{e}\right)}^{-1}\left(1-P_{Z}\left(1-S\right)\right)r_{d}$$

On the other hand, as derived from (13), the error at the motor side is formulated as
(18)$$e_{j}^{c1}=r_{d}-y_{j}^{m}=r_{d}-P_{M}v_{j}=Sr_{d}-P_{M}Su_{j}$$

Based on these expressions for the steady-state errors, we put forward the following proposition. This proposition underscores the equilibrium between errors and control inputs, not only on the table side but also on the motor side.

**Proposition** **1.**
*Assume that the DILC system in Figure 3 is stable. Then*
${\left\|\left(1-P_{Z}\right)r_{d}\right\|}_{2}\leq \;{\left\|e_{j}^{z1}\right\|}_{2}+{\left\|P_{z}\right\|}_{2}{\left\|e_{j}^{c1}\right\|}_{2}$*, where* 
${\left\|P\right\|}_{2}$ 
*denotes the *
2
*-norm of* 
P
*;*$u_{\infty }^{c}={\left(1-L_{u}+P_{Z}P_{M}SL_{e}\right)}^{-1}\left[C_{P}\left(1-L_{u}\right)+L_{e}\right]Sr_{d}-u_{\infty }$.


**Proof of Proposition** **1.**(a) According to (14),
(19)$$\begin{array}{ll} e_{j}^{z1} & =\left(1-P_{Z}\left(1-S\right)\right)r_{d}-P_{Z}P_{M}Su_{j}\\ & =P_{z}\left(Sr_{d}-P_{M}Su_{j}\right)+\left(1-P_{z}\right)r_{d}\\ & =P_{Z}e_{j}^{c1}+\left(1-P_{Z}\right)r_{d} \end{array}$$Then the result follows from the triangular inequality.(b) From Figure 4 and (17), the steady-state control input from the feedback controller is
(20)$$\begin{array}{l} u_{\infty }^{c}=C_{P}Sr_{d}-C_{P}P_{M}Su_{\infty }=C_{P}Sr_{d}-\left(1-S\right)u_{\infty }\\ =\left[C_{P}S-\left(1-S\right)L_{e}{\left(1-L_{u}+P_{Z}P_{M}SL_{e}\right)}^{-1}(1-P_{Z}\left(1-S\right))\right]r_{d}\\ ={\left(1-L_{u}+P_{Z}P_{M}SL_{e}\right)}^{-1}\left[C_{P}S\left(1-L_{u}\right)+\left(1-S\right)P_{Z}SL_{e}-\left(1-S\right)L_{e}(1-P_{Z}\left(1-S\right))\right]r_{d}\\ ={\left(1-L_{u}+P_{Z}P_{M}SL_{e}\right)}^{-1}\left[C_{P}\left(1-L_{u}\right)+L_{u}\right]Sr_{d}-u_{\infty } \end{array}$$□

Proposition 1a posits that both $e_{j}^{z1}$ and $e_{j}^{c1}$ are restricted by a lower bound, which is ${\left\|\left(1-P_{Z}\right)r_{d}\right\|}_{2}$. This suggests that for a specified $r_{d}$, they cannot simultaneously diminish to zero. Proposition 1b, on the other hand, illustrates inherent conflict between the feedback controller and DILC. To be specific, $u_{\infty }^{c}$ cancels out $u_{\infty }$, signifying that they counteract each other in the steady state.

This observation can be understood by examining the resonance modes illustrated in Figure 2. Structural vibrations, induced by large accelerations within the trajectory reference, cause relative motion between the motor and table sides. Concentrating solely on mitigating errors on one side inevitably leads to an amplification of errors on the opposite side, given the interconnected nature of these components. Consequently, the positioning accuracy attained is suboptimal.

In addition, we explore an extreme scenario where $w_{s}=0$ in (5). According to (7), $L_{u}=1$, and based on (16), $e_{\infty }^{z1}=0$. By employing Equations (17) and (20), it is determined that
(21)$$u_{\infty }={\left(P_{Z}P_{M}S\right)}^{-1}\left(1-P_{Z}\left(1-S\right)\right)r_{d}$$
and
(22)$$u_{\infty }^{c}={\left(P_{Z}P_{M}\right)}^{-1}r_{d}-u_{\infty }$$

For systems with rigid connections, that is, $P_{Z}=1$, Equation (22) indicates that $u_{\infty }^{c}=0$, implying that only DILC contributes to the control input in the steady state. However, when $P_{Z}\neq 1$, there occurs a cancellation effect between $u_{\infty }^{c}$ and $u_{\infty }$. Moreover, in a well-designed FBC system, $\left|S(\omega )\right|\ll 1$ for the frequencies within the control bandwidth. Consequently, according to (22), both $u_{\infty }$ and $u_{\infty }^{c}$ could become exceedingly large, potentially leading to the saturation of the hardware devices or introducing numerical complexities in the implementation of the control algorithm.

## 4. Proposed Approach and Analysis

As previously mentioned, direct application of DILC to flexible feed drive systems results in a conflict between the objectives of FBC and ILC. To mitigate this issue, this section aims to suppress both motor side errors and table side errors. This objective is achieved by refining the control compensation and reference inputs, leading to the development of a multivariable ILC (MILC) architecture. This strategy forms contribution C2 of this study. In addition, comprehensive insights into the stability analysis and the specific implementation of this architecture are thoroughly discussed. 

### 4.1. Multivariable ILC Design

The structure of the MILC is depicted in Figure 4, where $r_{j}$ and $u_{j}$ represent the reference update and the control compensation update, respectively. It is important to note that during the initial trial, $r_{0}=C_{y}r_{d}$ with the prefilter $C_{y}=P_{Z}^{-1}$, and $u_{0}=0$. Within this MILC framework, the calculations of $y_{j}^{m}$ and $y_{j}^{t}$ are as follows:(23)$$y_{j}^{m}=C_{P}P_{M}Sr_{j}+P_{M}Su_{j}$$
(24)$$y_{j}^{t}=C_{P}P_{M}P_{Z}Sr_{j}+P_{M}P_{z}Su_{j}$$

In this context, $e_{j}^{c2}$ signifies the motor side error utilized for the feedback controller within the MILC framework. This is expressed as
(25)$$e_{j}^{c2}=r_{j}-y_{j}^{m}$$

Furthermore, $e_{j}^{z2}$ representing the table side error, is the ultimate interest for control, and expressed as follows: (26)$$e_{j}^{z2}=r_{d}-y_{j}^{t}$$

The Equations (23)–(26) can be reorganized into matrix form as follows:(27)$$\begin{array}{ll} \left[\begin{array}{c} e_{j}^{c2}\\ e_{j}^{z2} \end{array}\right] & =\left[\begin{array}{c} 0\\ r_{d} \end{array}\right]-\left[\begin{array}{c} y_{j}^{m}-r_{j}\\ y_{j}^{t} \end{array}\right]\\ & =\left[\begin{array}{c} 0\\ r_{d} \end{array}\right]-\left[\begin{array}{cc} P_{M}S & -S\\ P_{M}P_{Z}S & C_{P}P_{M}P_{Z}S \end{array}\right]\left[\begin{array}{c} u_{j}\\ r_{j} \end{array}\right] \end{array}$$

Define the learning model of MILC as $H_{M}=\left[\begin{array}{cc} P_{M}S & -S\\ P_{M}P_{Z}S & C_{P}P_{M}P_{Z}S \end{array}\right]$; then
(28)$$\left[\begin{array}{c} e_{j}^{c2}\\ e_{j}^{z2} \end{array}\right]=\left[\begin{array}{c} 0\\ r_{d} \end{array}\right]-H_{M}\left[\begin{array}{c} u_{j}\\ r_{j} \end{array}\right]$$

The MILC update rule in lifted system format is as follows:(29)$$\left[\begin{array}{c} {\underline{u}}_{j+1}\\ {\underline{r}}_{j+1} \end{array}\right]=\underline{L_{u}}\left[\begin{array}{c} {\underline{u}}_{j}\\ {\underline{r}}_{j} \end{array}\right]+\underline{L_{e}}\left[\begin{array}{c} {\underline{e}}_{j}^{c2}\\ {\underline{e}}_{j}^{z2} \end{array}\right]$$

The purpose of this design is to effectively suppress both motor side FBC error $e_{j}^{c2}$ and table side performance error $e_{j}^{z2}$, while avoiding the issues associated with DILC, as highlighted in Proposition 1. Equation (29) implies that the MILC strategy is a two-DOF approach. 

This paper uses the following algorithm steps to demonstrate the algorithm process:
Step 1: Set $r_{j}=\frac{1}{P_{z}}r_{d},\;u_{j}=0$. Perform the tracking control task in real time with FBC only.Step 2: For j≥0, collect errors $e_{j}^{c2}$ and $e_{j}^{z2}$, and use Formula (9) to offline calculate the compensation for the next iteration, $r_{j+1}$ and $u_{j+1}$.Step 3: Perform the tracking control task in real time with reference $r_{j+1}$, and ILC compensation $u_{j+1}$.Step 4: Return to Step 2 and repeat until the desired number of iterations is achieved.


### 4.2. Implementation Aspects

Fundamentally, MILC functions as a multiple-input, multiple-output (MIMO) system, whose learning rule can be realized through the model-based NOILC detailed in Section 2 and Appendix A. Furthermore, the stability of MILC is guaranteed if the weighting matrices in (5) are chosen appropriately, as shown in Section 4.3. The precondition for efficient implementation of the NOILC method is to possess a state-space model that maps ILC learning inputs to the desired output, as outlined in Appendix A. To obtain this model, the blocks in Figure 4 are expressed in state-space representation, as follows:(30)$$C_{p}=\left[\begin{array}{cc} A_{c} & B_{c}\\ C_{c} & D_{c} \end{array}\right]$$
(31)$$P_{m}=\left[\begin{array}{cc} A_{m} & B_{m}\\ C_{m} & 0 \end{array}\right]$$
(32)$$P_{z}=\left[\begin{array}{cc} A_{z} & B_{z}\\ C_{z} & 0 \end{array}\right]$$
and they are assumed to be minimal realizations.

**Lemma** **1.***The state space representation of* $H_{M}$ *as depicted in Equation (28), specifically the transfer matrix from* $\left(u_{j},\;r_{j}\right)$ *to* $\left(-e_{j}^{c2},\;y_{j}^{t}\right),$ *is given by:*(33)$$H_{M}=\left[\begin{array}{cc} A & B\\ C & D \end{array}\right]$$
with
$$A=\left[\begin{array}{ccc} \begin{array}{c} A_{c}\\ B_{m}C_{c}\\ 0 \end{array} & \begin{array}{c} -B_{c}C_{m}\\ A_{m}-B_{m}D_{c}C_{m}\\ B_{z}C_{m} \end{array} & \begin{array}{c} 0\\ 0\\ A_{z} \end{array} \end{array}\right],\;B=\left[\begin{array}{cc} \begin{array}{c} 0\\ B_{m}\\ 0 \end{array} & \begin{array}{c} B_{c}\\ B_{m}D_{c}\\ 0 \end{array} \end{array}\right],\;C=\left[\begin{array}{c} \begin{array}{ccc} 0 & C_{m} & 0 \end{array}\\ \begin{array}{ccc} 0 & 0 & C_{z} \end{array} \end{array}\right],\;\mathrm{and}\;D=\left[\begin{array}{cc} 0 & -1\\ 0 & 0 \end{array}\right].$$
where boldface **0** denotes the zero matrix with appropriate dimension.

**Proof of Lemma** **1.**By systematically substituting the state updates and the output interconnection as outlined in (27), (30), (31), and (32), we arrive at the aforementioned representation. □

### 4.3. Stability and Convergence Analysis of MILC

In this subsection, monotonic convergence conditions of MILC are explored for the nominal plant model. 

**Theorem** **2.***(Monotonic convergence condition of MILC) In the NOLILC algorithm, where *$W_{q}≽0,W_{s}≽0$*, and *$W_{r}≽0$ 
*as defined in Definition 1, monotonic convergence for MILC is guaranteed for MILC learning laws (29) if *
$W_{q}$ 
*and* 
$W_{s}$ 
*are symmetric,* 
$W_{s}≻0$
*, and* 
$W_{r}=w_{r}I_{N}$
*, where* 
$w_{r}>0$ 
*and* 
$I_{N}$ 
*denotes an* 
N×N 
*identity matrix.*

**Proof of Theorem** **2.**According to (28) and (29), the trial dynamics of MILC can be expressed as
(34)$$\left[\begin{array}{c} {\underline{u}}_{j+1}\\ {\underline{r}}_{j+1} \end{array}\right]=\left({\underline{L}}_{u}-{\underline{L}}_{e}\underline{H_{M}}\right)\left[\begin{array}{c} {\underline{u}}_{j}\\ {\underline{r}}_{j} \end{array}\right]+{\underline{L}}_{e}\left[\begin{array}{c} 0\\ {\underline{r}}_{d} \end{array}\right]$$Monotonic convergence [33] of the MILC output requires that
(35)$${\left\|{\underline{L}}_{u}-{\underline{L}}_{e}\underline{H_{M}}\right\|}_{i2}<1$$
where ${\left\|\left(•\right)\right\|}_{i2}$ denotes the induced 2-norm a of matrix (•). For the NOILC design learning in (7) and (8), we can derive
(36)$${\left\|{\underline{L}}_{u}-{\underline{L}}_{e}\underline{H_{M}}\right\|}_{i2}={\left\|{\left({\underline{H_{M}}}^{T}W_{q}\underline{H_{M}}+W_{s}+W_{r}\right)}^{-1}W_{r}\right\|}_{i2}<1$$Given the conditions that $W_{q}≽0$, $W_{s}≻0$ and they are symmetric, we have $X={\underline{H_{M}}}^{T}W_{q}\underline{H_{M}}+W_{s}$ is symmetric and positive definite. In addition,
(37)$${\left({\underline{H_{M}}}^{T}W_{q}\underline{H_{M}}+W_{s}+W_{r}\right)}^{-1}W_{r}={\left(w_{r}^{-1}X+I_{N}\right)}^{-1}$$Let $\lambda_{i}(•)$ denote the i-th eigenvalue of (•). Since $w_{r}^{-1}X≻0$ and is symmetric, we have $\lambda_{i}\left(w_{r}^{-1}X+I_{N}\right)=\lambda_{i}\left(w_{r}^{-1}X\right)+1>1$ for i=1,⋯,N. Therefore, the following result is obtained:(38)$$\underset{i}{\max}\lambda_{i}{\left(w_{r}^{-1}X+I_{N}\right)}^{-1}<1$$Moreover, $w_{r}^{-1}X+I_{N}$ is symmetric. Thus
(39)$${\left\|{\left(w_{r}^{-1}X+I_{N}\right)}^{-1}\right\|}_{i2}=\underset{i}{\max}\lambda_{i}{\left(w_{r}^{-1}X+I_{N}\right)}^{-1}<1$$Therefore, inequality (36) holds and monotonic convergence of MILC is guaranteed. □

**Remark** **1.***Typically, the weighting matrices are chosen as* $\left(W_{q},\;W_{r},\;W_{s})≜(w_{q}I_{N},\;w_{r}I_{N},\;w_{s}I_{N}\right)$*, and the parameters* $w_{q}$*,* $w_{s}$*, and* $w_{r}$ *are commonly set as* $w_{q}>0$*,* $w_{s}>0$*, and* $w_{r}>0$.

**Remark** **2.**
*The sufficient condition for MC outlined in Theorem 2 is a general condition that is also applicable to the NOILC design of DILC.*


As stated in Proposition 1a, in DILC, $e_{j}^{c1}$ and $e_{j}^{z1}$ cannot be zero simultaneously. In contrast, MILC operates differently. Assuming that Theorem 2 holds, the conditions under which $e_{j}^{c2}$ and $e_{j}^{z2}$ can simultaneously be zero are outlined in the following remark. This demonstrates that MILC can effectively resolve the conflict between FBC and ILC.

**Remark** **3.***Given that $e_{j}^{z2}$* *= 0, it follows that $y_{j}^{t}=r_{d}$. Consequently, $y_{j}^{m}=\frac{1}{P_{z}}r_{d}$, and, since $e_{j}^{m2}=0$, we deduce $r_{j}^{*}=\frac{1}{P_{z}}r_{d}$. Therefore, $u_{j}^{*}=\frac{1}{P_{z}P_{m}}r_{d}$*.

## 5. Experimental Results

### 5.1. Experimental Setup

As shown in Figure 5, a biaxial motion stage, manufactured by Tongtai Machine & Tool Co., Ltd., Kaohsiung, Taiwan, is employed to assess the tracking performance of MILC. This stage is driven by two servo motors from Shihlin Electric & Engineering Co., Taipei, Taiwan. Each motor is equipped with a rotary encoder for a semi-closed position loop, providing a resolution of 52 μrad per pulse. The ball screw pitch lengths are 10 mm and 12 mm for the X and Z axes, respectively. Additionally, every axis is equipped with a linear encoder to measure the table position, offering a resolution of 0.25 μm per pulse. The actuators of both motors are controlled by a National Instruments cRIO 9035 embedded system with 4 kHz sampling frequency, and it offers a 16-bit resolution of analog output. The National Instruments cRIO 9035, made in Austin, TX, USA, ensures precise control and measurement. Since the two axes are independent, we apply the control laws to the *X* axis for experimental verification. 

The servo motor for each axis employs the commonly used three-loop control architecture, as depicted in Figure 6. This control structure comprises an outer position loop, an intermediate velocity loop, and an innermost current loop.

In this study, the closed current control loop, as delineated by the dashed box in Figure 6, is regarded as the plant. Within this setup, the torque command serves as the input and the velocity as the output. Enveloping the plant is the velocity control loop and subsequently the position control loop, both operating at a sampling rate of f = 4 kHz.

The discrete-time TF from the torque command to the velocity of the *X*-axis is represented by $P_{x}(z)$, while the TF from the motor position to the table side position is denoted as $P_{z}$(z). Both TFs have been identified through the analysis of experimental data. The magnitude responses of the TFs $P_{x}\left(z\right)$ and $P_{z}(z)$ are depicted in Figure 7.

In this framework, $C_{P}(z)$ denotes the position loop feedback controller, $C_{v}(z)$ denotes the velocity feedback controller, and $C_{i}(z)$ encapsulates the combination of the current loop controller and its corresponding filter [2]. The position loop controller is configured as a proportional (P) controller, and the velocity controller is designed as a proportional-integral (PI) controller. These controllers are tuned by using the PID tuning toolbox in MATLAB and digitally executed within the CRIO 9035 embedded system. The expressions for $C_{p}(z)$, $C_{v}(z)$, and a backward type integrator $G_{i}(z)$ are as follows:(40)$$C_{P}\left(z\right)=K_{pp}$$
(41)$$C_{v}\left(z\right)=K_{vp}+K_{vi}\frac{T_{s}z}{z-1},\;G_{i}\left(z\right)=\frac{T_{s}z}{z-1}$$
where $K_{vp}=0.216$, $K_{vi}=8.18$, $K_{pp}=180$, $T_{s}=1/f.$
(42)$$P_{M}\left(z\right)=\frac{C_{v}\left(z\right)P_{x}(z)}{1+C_{v}\left(z\right)P_{x}(z)}G_{i}(z)$$

By applying the controllers (38) and (39), the position closed loop possesses a bandwidth (−3 dB) of 50 Hz.

To demonstrate the significant performance improvement of MILC, DILC is also implemented by the norm-optimal approach for comparison purposes. The state-space representation for DILC implementations is provided in Appendix B. 

### 5.2. Experimental Results

The effectiveness of the proposed MILC is evaluated in comparison with DILC through experiments on the motion stage. The desired trajectory $r_{d}$ (illustrated in Figure 8) consists of a sequence of smoothed step references, and every step reference is designed using an S-curve velocity profile. This trajectory, encompassing a wide range of accelerations, is apt for evaluating the characteristics of a dynamic system in high-throughput manufacturing scenarios. Due to practical considerations of model mismatch [34], the performance weights in (5) are set as $w_{q}=2\times 10^{6}$, $w_{r}=10^{-4}$, and $w_{s}=0.2\times 10^{2}$ for both MILC and DILC schemes. Furthermore, feedforward friction compensation is effectively implemented by a look-up table, which is based on the result gathered from preliminary friction identified experiments. To prevent instantaneous fluctuations in friction compensation due to measurement noise in low-speed regions, the reference velocity is utilized as the input for the look-up table.

The experimental results are depicted in Figure 9 and Figure 10. The performance is evaluated using the root mean squared (RMS) and maximum (MAX) errors in every iteration. After repeated iterations, the relationship between iteration index and RMS of the tracking error is shown in Figure 9. With DILC, once the motor side error decreases to a certain level, further reductions are not observed despite an increase in iterations. In contrast, the proposed MILC successfully reduces errors at both the motor and table sides to a much smaller range. It should be noted that the proposed scheme is different from DILC because it iteratively corrects the reference input and control compensation, aligning the objectives of FBC and ILC for more rapid and effective convergence of table side errors.

Additionally, a comparison of the two approaches is made using the table side tracking error ($e_{j}^{zi},i=1,2$) in the 10-th iteration, as shown in Figure 10 and summarized in Table 1. In sections with high acceleration, such as near 0.57 s, 0.59 s, and 0.61 s as highlighted in the magnified view, the tracking performance of DILC significantly worsens, but MILC demonstrates a notable performance improvement. For the proposed MILC scheme, errors at both the motor side and table side can be simultaneously suppressed, and outstanding compensation performance can be achieved., i.e., MILC achieves a MAX error of 2.06 µm, which is approximately a 42% improvement compared to DILC. As inferred from Figure 2 and Figure 7, due to the presence of large accelerations, resonance modes of $P_{x}$ and $P_{Z}$, are triggered, resulting in relative motion between the motor side and the table side. Controlling both sides is necessary to achieve high-speed and high-precision motion control.

Figure 11 displays the control input from the feedback controller in the 10th trial for DILC and MILC. Due to the cancellation between $u_{j}^{c}$ and $u_{j}$, DILC exhibits a substantial control input from the feedback controller. Conversely, MILC, by simultaneously tunning the reference input and the control compensation, maintains a small control input from the feedback controller in the steady state. This result is attributed to the consistent objectives between FBC and MILC to jointly suppress errors at both the motor and table sides, while there are conflicts in control objectives between FBC and DILC as illustrated in Proposition 1b.

## 6. Conclusions

This paper presents a novel multivariable ILC strategy designed to improve the dynamic positioning accuracy of flexible feed drive motion systems by simultaneously reducing errors at both the motor and table sides. The approach utilizes a two-DOF control framework, combining iterative reference shaping with control compensation updates, effectively balancing the objectives of FBC and ILC. This development is driven by the need to address relative movements caused by the flexible couplings between the motor and table sides. These movements are prompted by various disturbances, including closed-loop dynamics, lead error, structural vibrations, and thermal error. The proposed method has been validated on an industrial CNC machine tool, demonstrating improvements in dynamic tracking errors of up to 42% (as seen in Figure 10 and Table 1). Overall, this MILC can be implemented without the need for altering the servo controller. It provides a convenient approach by modifying the reference trajectory and control compensation. However, injecting the control compensation into the internal controller may encounter some difficulties in certain closed commercial controllers. The study in this paper shows that MILC has appreciable application prospects for precision/ultra-precision systems with flexible feed drives to meet extreme motion accuracy requirements. Future research could further explore the real-time implementation of these advanced control schemes on standard industrial controllers.

## Figures and Tables

**Figure 1 sensors-24-03536-f001:**
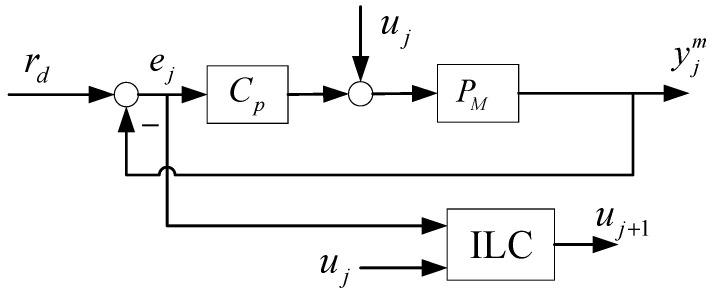
Standard ILC setup.

**Figure 2 sensors-24-03536-f002:**
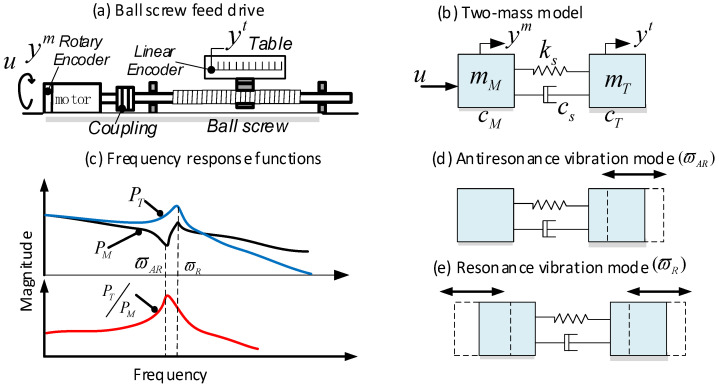
Dynamics of flexible ball screw feed drives. (Redrawn based on Figure 2 of [32]).

**Figure 3 sensors-24-03536-f003:**
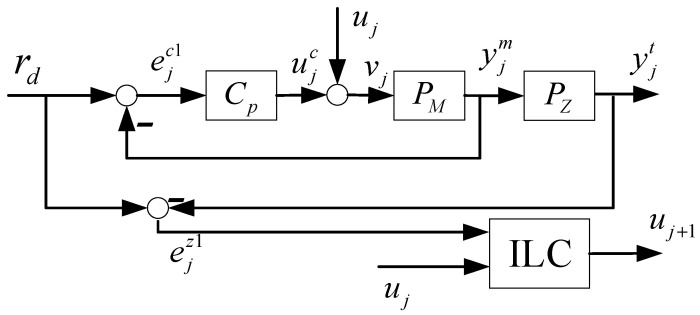
ILC directly applied to flexible structure (DILC).

**Figure 4 sensors-24-03536-f004:**
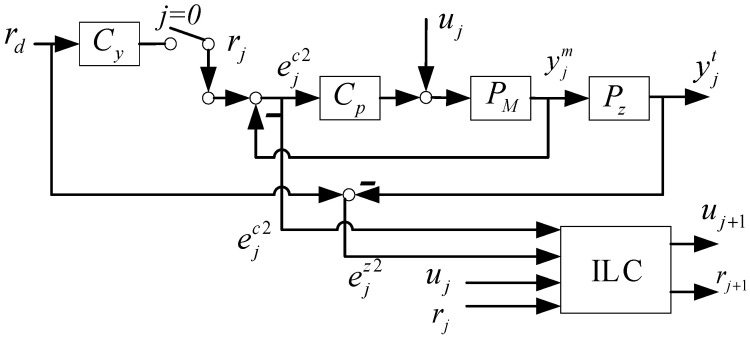
Two-DOF based MILC configuration.

**Figure 5 sensors-24-03536-f005:**
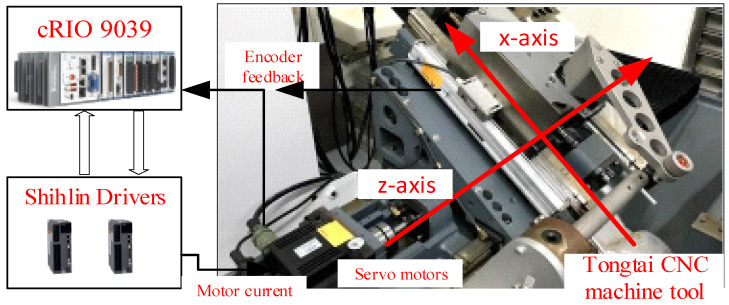
CNC machine tool motion system.

**Figure 6 sensors-24-03536-f006:**
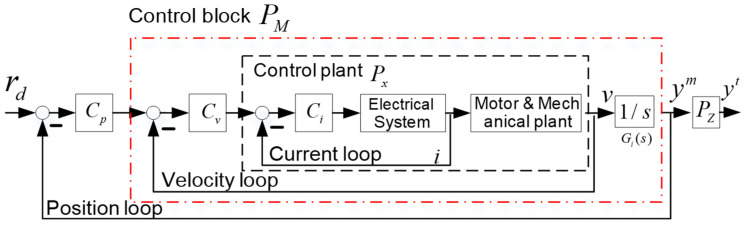
Servo motor cascaded three-loop structure.

**Figure 7 sensors-24-03536-f007:**
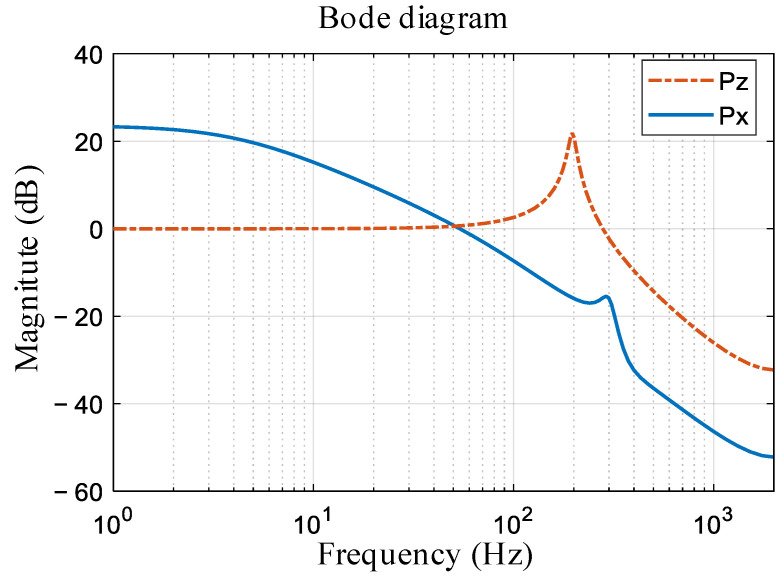
Open loop frequency response functions of nominal *Px* and *Pz*.

**Figure 8 sensors-24-03536-f008:**
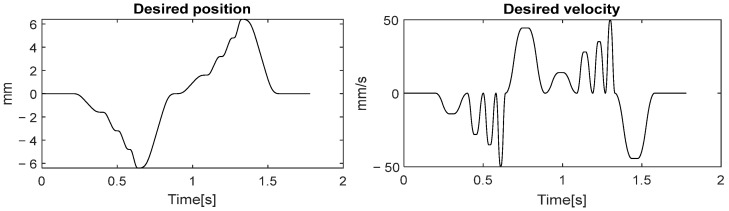
Desired trajectory is a forward and backward movement.

**Figure 9 sensors-24-03536-f009:**
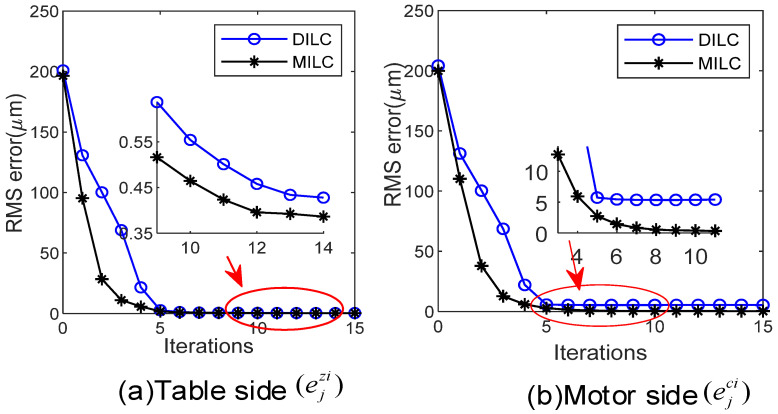
RMS of tracking error by DILC and MILC.

**Figure 10 sensors-24-03536-f010:**
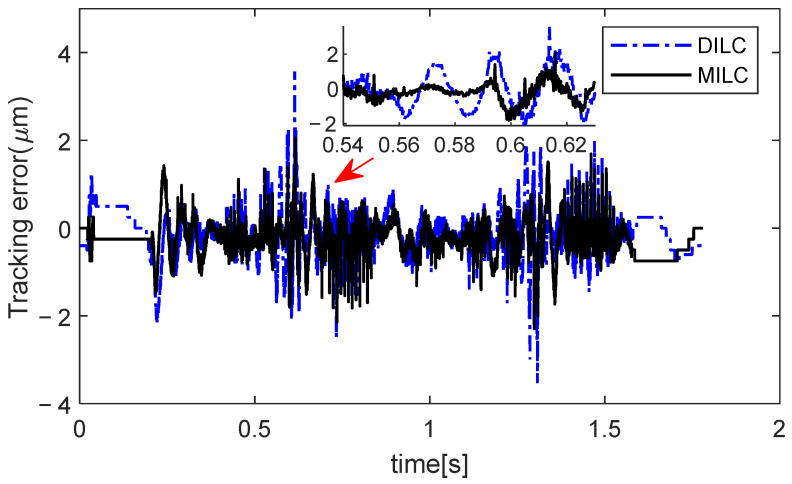
Comparison of $e_{j}^{zi}$ in 10-th trial (*j* = 10) for different ILCs.

**Figure 11 sensors-24-03536-f011:**
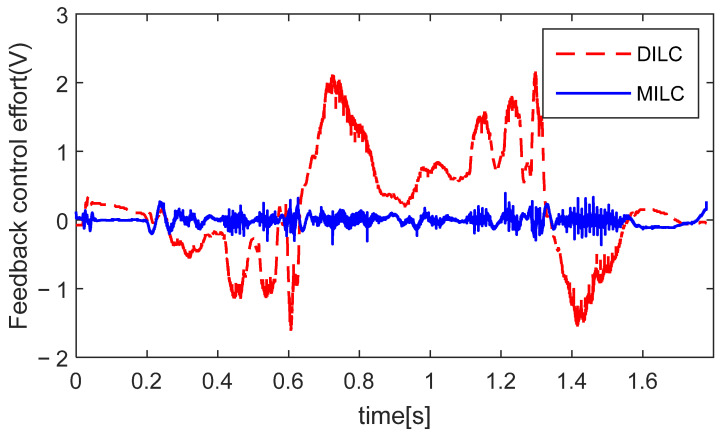
FBC effort $u^{c}$ in 10-th trial for different ILCs.

**Table 1 sensors-24-03536-t001:** Performance in different approaches (μm, [The symbol “*” indicates the best result under the current metrics]).

Method	$$\mathrm{RMS}\;({\underline{e}}_{0}^{zi})$$	$$\mathrm{RMS}\;({\underline{e}}_{0}^{ci})$$	$$\mathrm{RMS}\;({\underline{e}}_{10}^{zi})$$	$$\mathrm{RMS}\;({\underline{e}}_{10}^{ci})$$	$$\mathrm{MAX}\;({\underline{e}}_{10}^{zi})$$
DILC*_i_*_=1_	201.00	204.21	0.55	5.39	3.57
MILC*_i_*_=2_	196.46 *	199.81 *	0.46 *	0.40 *	2.06 *

## Data Availability

Data are contained within the article.

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
