# Peer review of "Multivariable Iterative Learning Control Design for Precision Control of Flexible Feed Drives"

_sensors, 2024, doi:10.3390/s24113536_

Round 1

Reviewer 1 Report

Comments and Suggestions for Authors

Authors present a multivariable iterative learning control method for flexible feed drive systems, which enhances dynamic positioning accuracy. The method employs error data from both the motor and table sides, increasing precision by injecting compensation commands into both the reference trajectory and control command. The proposed method reduce tracking errors in the table side and was empirically validated using a biaxial CNC machine tool.

The topic is actual and important.

The paper is generally well written.

The English of the article can be probably corrected.

Detailed remarks:

Line 26 and many other: “manipulators[1]” => “manipulators [1]” (please insert single space before “[“)

Line 459 and many other: “f=4kHz” => “f = 4 kHz”

Line 507 – please improve visibility of parts of Fig.9

Author Response

Comments 1: Line 26 and many other: “manipulators[1]” => “manipulators [1]” (please insert single space before “[“

Response 1: Thank you for pointing out the error. We have corrected this throughout the document. Thank you for your valuable feedback.

Comments 2: Line 459 and many other: “f=4kHz” => “f = 4 kHz”

Response 2:  Thank you for highlighting this issue. We have made the necessary corrections throughout the text. Thank you for your valuable feedback.

Comments 3: Line 507 – please improve visibility of parts of Fig.9

Response 3: Thank you for pointing this out. We have enlarged the relevant parts of Figure 9 to improve visibility.

Reviewer 2 Report

Comments and Suggestions for Authors

Reviewer 3 Report

Comments and Suggestions for Authors

Dear Authors
The manuscript is related to an issue of control system design for flexible feed drive systems. The work included a proposal for a multivariable iterative learning control (MILC) method to enhance dynamic positioning accuracy. The results were compared to the traditional iterative learning control directly applied to flexible structure (DILC). The issue is crucial in systems where disturbances such as structural vibrations or thermal deformations occur.
The efficiency of the proposed MILC system was validated experimentally, which is especially valuable.
The article is well-formed and fits well with the journal's subject matter. The title is appropriate. The abstract is clear. However, I have several suggestions that should be addressed:

Introduction:
1. The reference list includes 38 positions. However, there are no references from Sensors. You could consider adding one or two suitable references to better justify your subject as appropriate for this journal.

Methodology/research/conclusions:
2. The test bench should be described in more detail, including the ranges and accuracies of the used measuring and data acquisition equipment.
3. Clear specifications of both advantages and limitations of your proposal should be provided.
4. Conclusions are very brief and should be extended. You just repeated your work and the results you obtained. However, you may also formulate more general conclusions from your research.

Editorial:
5. You refer to section 4.2 inside section 4.2 (line 390).
6. Figs 7,8,9,10,11 could be enlarged for better visibility.

Regards,

Round 2

Reviewer 2 Report

Comments and Suggestions for Authors

I have no further comments.